# MODEL IMITATION FOR MODEL-BASED REINFORCEMENT LEARNING

## ABSTRACT

Model-based reinforcement learning (MBRL) aims to learn a dynamic model to reduce the number of interactions with real-world environments. However, due to estimation error, rollouts in the learned model, especially those of long horizons, fail to match the ones in real-world environments. This mismatching has seriously impacted the sample complexity of MBRL. The phenomenon can be attributed to the fact that previous works employ supervised learning to learn the one-step transition models, which has inherent difficulty ensuring the matching of distributions from multi-step rollouts. Based on the claim, we propose to learn the synthesized model by matching the distributions of multi-step rollouts sampled from the synthesized model and the real ones via WGAN. We theoretically show that matching the two can minimize the difference of cumulative rewards between the real transition and the learned one. Our experiments also show that the proposed *model imitation* method outperforms the state-of-the-art in terms of sample complexity and average return.

## 1 INTRODUCTION

Reinforcement learning (RL) has become of great interest because plenty of real-world problems can be modeled as a sequential decision-making problem. Model-free reinforcement learning (MFRL) is favored by its capability of learning complex tasks when interactions with environments are cheap. However, in the majority of real-world problems, such as autonomous driving, interactions are extremely costly, thus MFRL becomes infeasible. One critique about MFRL is that it does not fully exploit past queries over the environment, and this motivates us to consider the model-based reinforcement learning (MBRL). In addition to learning an agent policy, MBRL also uses the queries to learn the dynamics of the environment that our agent is interacting with. If the learned dynamic is accurate enough, the agent can acquire the desired skill by simply interacting with the simulated environment, so that the number of samples to collect in the real world can be greatly reduced. As a result, MBRL has become one of the possible solutions to reduce the number of samples required to learn an optimal policy.

Most previous works of MBRL adopt supervised learning with $\ell_2$-based errors (Luo et al., 2019; Kurutach et al., 2018; Clavera et al., 2018) or maximum likelihood (Janner et al., 2019), to obtain an environment model that synthesizes real transitions. These non-trivial developments imply that optimizing a policy on a synthesized environment is a challenging task. Because the estimation error of the model accumulates as the trajectory grows, it is hard to train a policy on a long synthesized trajectory. On the other hand, training on short trajectories makes the policy short-sighted. This issue is known as the planning horizon dilemma (Wang et al., 2019). As a result, despite having a strong intuition at first sight, MBRL has to be designed meticulously.

Intuitively, we would like to learn a transition model in a way that it can reproduce the trajectories that have been generated in the real world. Since the attained trajectories are sampled according to a certain policy, directly employing supervised learning may not necessarily lead to the mentioned result especially when the policy is stochastic. The resemblance in trajectories matters because we estimate policy gradient by generating rollouts; however, the one-step model learning adopted by many MBRL methods do not guarantee this. Some previous works propose multi-step training (Luo et al., 2019; Asadi et al., 2019; Talvitie, 2017); however, experiments show that model learning fails to benefit much from the multi-step loss. We attribute this outcome to the essence of super-

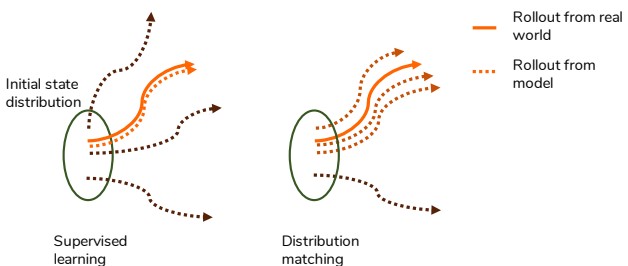

Figure 1: Distribution matching enables the learned transition to generate similar rollouts to the real ones even when the policy is stochastic or the initial states are close. On the other hand, training with supervised learning does not ensure rollout similarity and the resulting policy gradient may be inaccurate. This figure considers a fixed policy sampling in the real world and a transition model.

vised learning, which elementally preserves only one-step transition and the similarity between real trajectories and the synthesized ones cannot be guaranteed.

In this work, we propose to learn the transition model via distribution matching. Specifically, we use WGAN (Arjovsky et al., 2017) to match the distributions of state-action-next-state triple $(s, a, s')$ in real/learned models so that the agent policy can generate similar trajectories when interacting with either the true transition or the learned transition. Figure 1 illustrates the difference between methods based on supervised learning and distribution matching. Different from the ensemble methods proposed in previous works, our method is capable of generalizing to unseen transitions with only *one* dynamic model because merely incorporating multiple models does not alter the essence that one-step (or few-step) supervised learning fails to imitate the distribution of multi-step rollouts.

Concretely, we gather some transitions in the real world according to a policy. To learn the real transition $T$, we then sample fake transitions from our synthesized model $T'$ with the same policy. The synthesized model serves as the generator in the WGAN framework and there is a critic that discriminates the two transition data. We update the generator and the critic alternatively until the synthesized data cannot be distinguished from the real one, which we will show later that it gives $T \to T'$ theoretically.

Our contributions are summarized below:

- We propose an MBRL method called model imitation (MI), which enforces the learned transition model to generate similar rollouts to the real one so that policy gradient is accurate;
- We theoretically show that the transition can be learned by MI in the sense that $T \to T'$ by consistency and the difference in cumulative rewards can be bounded by the training error of WGAN;
- To stabilize model learning, we deduce guarantee for our sampling technique and investigate training across WGANs;
- We experimentally show that MI is more sample efficient than state-of-the-art MBRL and MFRL methods and outperforms them on four standard tasks.

## 2 RELATED WORK

In this section, we introduce our motivation inspired by learning from demonstration (LfD) (Schaal, 1997) and give a brief survey of MBRL methods.

### 2.1 LEARNING FROM DEMONSTRATION

A straightforward approach to LfD is to leverage behavior cloning (BC), which reduces LfD to a supervised learning problem. Even though learning a policy via BC is time-efficient, it cannot imitate a policy without sufficient demonstration because the error may accumulate without the guidance of expert (Ross et al., 2011). Generative Adversarial Imitation Learning (GAIL) (Ho & Ermon, 2016) is another state-of-the-art LfD method that learns an optimal policy by utilizing

generative adversarial training to match occupancy measure (Syed et al., 2008). GAIL learns an optimal policy by matching the distribution of the trajectories generated from an agent policy with the distribution of the given demonstration. Ho & Ermon (2016) shows that the two distributions match if and only if the agent has learned the optimal policy. One of the advantages of GAIL is that it only requires a small amount of demonstration data to obtain an optimal policy but it requires a considerable number of interactions with environments for the generative adversarial training to converge.

Our intuition is that transition learning (TL) is similar to learning from demonstration (LfD) by exchanging the roles of transition and policy. In LfD, trajectories sampled from a fixed transition are given, and the goal is to learn a policy. On the other hand, in TL, trajectories sampled from a fixed policy are given, and we would like to imitate the underlying transition. That being said, from LfD to TL, we interchange the roles of the policy and the transition. It is therefore tempting to study the counterpart of GAIL in TL; i.e., *learning the transition by distribution matching*. Fortunately, by doing so, the pros of GAIL remain while the cons are insubstantial in MBRL because sampling with the learned model is considered to be much cheaper than sampling in the real one. That GAIL learns a better policy than what BC does suggests that distribution matching possesses the potential to learn a better transition than supervised learning.

## 2.2 MODEL-BASED REINFORCEMENT LEARNING

For deterministic transition, $\ell_{@}$-based error is usually utilized to learn the transition model. Nagabandi et al. (2018), an approach that uses supervised learning with mean-squared error as its objective, is shown to perform well under fine-tuning. To alleviate model bias, some previous works adopt ensembles (Kurutach et al., 2018; Buckman et al., 2018), where multiple transition models with different initialization are trained at the same time. In a slightly more complicated manner, Clavera et al. (2018) utilizes meta-learning to gather information from multiple models. Lastly, on the theoretical side, SLBO (Luo et al., 2019) is the first algorithm that develops from solid theoretical properties for model-based deep RL via a joint model-policy optimization framework.

For the stochastic transition, maximum likelihood estimator or moment matching are natural ways to learn a synthesized transition, which is usually modeled by the Gaussian distribution. Following this idea, Gaussian process (Kupcsik et al., 2013; Deisenroth et al., 2015) and Gaussian process with model predictive control (Kamthe & Deisenroth, 2017) are introduced as an uncertainty-aware version of MBRL. Similar to the deterministic case, to mitigate model bias and foster stability, an ensemble method for probabilistic networks (Chua et al., 2018) is also studied. An important distinction between training a deterministic or stochastic transition is that although the stochastic transition can model the noise hidden within the real world, the stochastic model may also induce instability if the true transition is deterministic. This is a potential reason why an ensemble of models is adopted to reduce variance.

## 3 BACKGROUND

### 3.1 REINFORCEMENT LEARNING

We consider the standard Markov Decision Process (MDP) (Sutton & Barto, 1998). MDP is represented by a tuple $\langle S, A, T, r, \gamma \rangle$, where $S$ is the state space, $A$ is the action space, $T(s_{t+1}|s_t, a_t)$ is the transition density of state $s_{t+1}$ at time step $t + 1$ given action $a_t$ made under state $s_t$, $r(s, a)$ is the reward function, and $\gamma \in (0, 1)$ is the discount factor.

A stochastic policy $\pi(a|s)$ is a density of action $a$ given state $s$. Let the initial state distribution be $\alpha$. The performance of the triple $(\alpha, \pi, T)$ is evaluated in the expectation of the cumulative reward in the $\gamma$-discounted infinite horizon setting:

$$R(\alpha, \pi, T) = \mathbb{E}\left[\sum_{t=0}^{\infty} \gamma^t r(s_t, a_t) \Big| \alpha, \pi, T\right] = \mathbb{E}\left[\sum_{t=0}^{H-1} r(s_t, a_t) \Big| \alpha, \pi, T\right]. \tag{1}$$

Equivalently, $R(\alpha, \pi, T)$ is the expected cumulative rewards in a length-$H$ trajectory $\{s_t, a_t\}_{t=0}^{H-1}$ generated by $(\alpha, \pi, T)$ with $H \sim \text{Geometric}(1 - \gamma)$. When $\alpha$ and $T$ are fixed, $R(\cdot)$ becomes a

function that only depends on $\pi$, and reinforcement learning algorithms (Sutton & Barto, 1998) aim to find a policy $\pi$ to maximize $R(\pi)$.

## 3.2 OCCUPANCY MEASURE

Given initial state distribution $\alpha(s)$, policy $\pi(a|s)$ and transition $T(s'|s,a)$, the normalized occupancy measure $\rho_T^{\alpha,\pi}(s,a)$ generated by $(\alpha, \pi, T)$ is defined as

$$\rho_T^{\alpha,\pi}(s,a) = \sum_{t=0}^{\infty}(1-\gamma)\gamma^t \mathbb{P}(s_t = s, a_t = a|\alpha, \pi, T) = (1-\gamma)\sum_{t=0}^{H-1} \mathbb{P}(s_t = s, a_t = a|\alpha, \pi, T), \quad (2)$$

where $\mathbb{P}(\cdot)$ is the probability measure and will be replaced by a density function if $\mathcal{S}$ or $\mathcal{A}$ is continuous. Intuitively, $\rho_T^{\alpha,\pi}(s,a)$ is a distribution of $(s,a)$ in a length-$H$ trajectory $\{s_t, a_t\}_{t=0}^{H-1}$ with $H \sim \text{Geometric}(1-\gamma)$ following the laws of $(\alpha, \pi, T)$. From Syed et al. (2008), the relation between $\rho_T^{\alpha,\pi}$ and $(\alpha, \pi, T)$ is characterized by the Bellman flow constraint. Specifically, $x = \rho_T^{\alpha,\pi}$ as defined in Eq. 2 is the unique solution to:

$$x(s,a) = \pi(a|s)\Big[(1-\gamma)\alpha(s) + \gamma \int x(s',a')T(s|s',a')ds'da'\Big], \quad x(s,a) \geq 0. \quad (3)$$

In addition, Theorem 2 of Syed et al. (2008) gives that $\pi(a|s)$ and $\rho_T^{\alpha,\pi}(s,a)$ have an one-to-one correspondence with $\alpha(s)$ and $T(s'|s,a)$ fixed; i.e., $\pi(a|s) \triangleq \frac{\rho(s,a)}{\int \rho(s,a)da}$ is the only policy whose occupancy measure is $\rho$.

With the occupancy measure, the cumulative reward Eq. 1 can be represented as

$$R(\alpha, \pi, T) = \mathbb{E}_{(s,a)\sim\rho_T^{\alpha,\pi}}[r(s,a)]/(1-\gamma). \quad (4)$$

The goal of maximizing the cumulative reward can then be achieved by adjusting $\rho_T^{\alpha,\pi}$, and this motivates us to adopt distribution matching approaches like WGAN (Arjovsky et al., 2017) to learn a transition model.

## 4 THEORETICAL ANALYSIS FOR WGAN

In this section, we present a consistency result and error bounds for WGAN (Arjovsky et al., 2017). All proofs of the following theorems and lemmas can be found in Appendix A.

In the setting of MBRL, the training objective for WGAN is

$$\min_{T'} \max_{\|f\|_{\text{Lip}} \leq 1} \mathbb{E}_{(s,a)\sim\rho_T^{\alpha,\pi}, \, s'\sim T(\cdot|s,a)}[f(s,a,s')] - \mathbb{E}_{(s,a)\sim\rho_{T'}^{\alpha,\pi}, \, s'\sim T'(\cdot|s,a)}[f(s,a,s')]. \quad (5)$$

By Kantorovich-Rubinstein duality (Villani, 2008), the optimal value of the inner maximization is exactly $W_1(p(s,a,s')||p'(s,a,s'))$ where $p(s,a,s') = \rho_T^{\alpha,\pi}(s,a)T(s'|s,a)$ is the discounted distribution of $(s,a,s')$. Thus, by minimizing over the choice of $T'$, we are essentially finding $p'$ that minimizes $W_1(p(s,a,s')||p'(s,a,s'))$, which gives the consistency result.

**Proposition 1** (Consistency for WGAN). *Let $T$ and $T'$ be the true and synthesized transitions respectively. If WGAN is trained to its optimal point, we have*

$$T(s'|s,a) = T'(s'|s,a), \, \forall(s,a) \in \text{Supp}(\rho_T^{\alpha,\pi}),$$

*where $\text{Supp}(\rho_T^{\alpha,\pi})$ is the support of $\rho_T^{\alpha,\pi}$.*

The support constraint is inevitable because the training data is sampled from $\rho_T$ and guaranteeing anything beyond it can be difficult. Still, we will empirically show that the support constraint is not an issue in our experiments because the performance boosts up in the beginning, indicating that $\text{Supp}(\rho_T^{\alpha,\pi})$ may be large enough initially.

Now that training with WGAN gives a consistent estimate of the true transition, it is sensible to train a synthesized transition upon it. However, the consistency result is too restrictive as it only discusses the optimal case. The next step is to analyze the non-optimal situation and observe how the cumulative reward deviates w.r.t. the training error.

**Theorem 1** (Error Bound for WGAN). *Let $\rho_T^{\alpha,\pi}(s,a)$, $\rho_{T'}^{\alpha,\pi}(s,a)$ be the normalized occupancy measures generated by the true transition $T$ and the synthesized one $T'$. If the reward function is $L_r$-Lipschitz and the training error of WGAN is $\epsilon$, we have $|R(\pi,T) - R(\pi,T')| \leq \epsilon L_r/(1-\gamma)$.*

Theorem 1 indicates that if WGAN is trained properly, i.e., having small $\epsilon$, the cumulative reward on the synthesized trajectory will be close to that on the true trajectory. As MBRL aims to train a policy on the synthesized trajectory, the accuracy of the cumulative reward over the synthesized trajectory is thus the *bottleneck*. Theorem 1 also implies that WGAN's error is linear to the (expected) length of the trajectory $(1-\gamma)^{-1}$. This is a sharp contrast to the error bounds in most RL literature, as the dependency on the trajectory length is usually quadratic (Syed & Schapire, 2010; Ross et al., 2011), or of an even higher order. Since WGAN gives us a better estimation of the cumulative reward in the learned model, the policy update becomes more accurate.

## 5 MODEL IMITATION FOR MODEL-BASED REINFORCEMENT LEARNING

In this section, we present a practical MBRL method called model imitation (MI) that incorporates the transition learning mentioned in Section 4.

### 5.1 SAMPLING TECHNIQUE FOR TRANSITION LEARNING

Due to the long-term digression, it is hard to train the WGAN directly from a long synthesized trajectory. To tackle this issue, we use the synthesized transition $T'$ to sample $N$ short trajectories with initial states sampled from the true trajectory.

To analyze this sampling technique, let $\beta < \gamma$ be the discount factor of the short trajectories so that the expected length is $\mathbb{E}[L] = (1-\beta)^{-1}$. To simplify the notation, let $\rho_{T'}^{\beta}$, $\hat{\rho}_T^{\beta}$, $\rho_T^{\beta}$, $\rho_T$ be the normalized occupancy measures of synthesized short trajectories, empirical true short trajectories, true short trajectories and the true long trajectories. Notice both $\rho_{T'}^{\beta}$ and $\rho_T^{\beta}$ are generated from the same initial distribution $\rho_T$. The 1-Wasserstein distance can be bounded by

$$W_1(\rho_{T'}^{\beta}||\rho_T) \leq W_1(\rho_{T'}^{\beta}||\hat{\rho}_T^{\beta}) + W_1(\hat{\rho}_T^{\beta}||\rho_T^{\beta}) + W_1(\rho_T^{\beta}||\rho_T).$$

$W_1(\rho_{T'}^{\beta}||\hat{\rho}_T^{\beta})$ is upper bounded by the training error of WGAN on short trajectories, which can be small empirically because the short ones are easier to imitate. $W_1(\hat{\rho}_T^{\beta}||\rho_T^{\beta}) = \mathbb{E}_L[O((NL)^{-1/d})] = O(((1-\beta)/N)^{1/d}/\beta)$ by Canas & Rosasco (2012) and Lemma 1, where $d$ is the dimension of $(s,a)$. $W_1(\rho_T^{\beta}||\rho_T) \leq \text{diam}(\mathcal{S} \times \mathcal{A})(1-\gamma)\beta/(\gamma-\beta)$ by Lemma 2 and $W_1 \leq D_{TV}\text{diam}(\mathcal{S} \times \mathcal{A})$ (Gibbs & Su, 2002), where $\text{diam}(\cdot)$ is the diameter. The second term encourages $\beta$ to be large while the third term does the opposite. Besides, $\beta$ need not be large if $N$ is large enough; in practice we may sample $N$ short trajectories to reduce the error from $W_1(\rho_{T'}||\rho_T)$ to $W_1(\rho_{T'}^{\beta}||\rho_T)$. Finally, since $\rho_{T'}^{\beta}$ is the occupancy measure we train on, from the proof of Theorem 1 we deduce that

$$|R(\pi,T) - R(\pi,T')| \leq W_1(\rho_{T'}^{\beta}||\rho_T)L_r/(1-\gamma).$$

Thus, WGAN may perform better under this sampling technique.

### 5.2 EMPIRICAL TRANSITION LEARNING

To learn the real transition based on the occupancy measure matching mentioned in Section 4, we employ a transition learning scheme by aligning the distribution of $(s,a,s')$ between the real and the learned environments. Inspired by how GAIL (Ho & Ermon, 2016) learns to align $(s,a)$ via solving an MDP with rewards extracted from a discriminator, we formulate an MDP with rewards from a discriminator over $(s,a,s')$. Specifically, the WGAN critic $f(s,a,s')$ in Eq. 5 is used as the (psuedo) rewards $r(s,a,s')$ of our MDP. Interestingly, there is a duality between GAIL and our transition learning: for GAIL, the transition is fixed and the objective is to train a policy to maximize the cumulative pseudo rewards, while for our transition learning, the policy is fixed and the objective is to train a synthesized transition to maximize the cumulative pseudo rewards.

In practice, since the policy is updated alternatively with the synthesized model, we are required to train a number of WGANs along with the change of the policy. Although the generators across

WGANs correspond to the same transition and can be similar, we observe that WGAN may get stuck at a local optimum when we switch from one WGAN training to another. The reason is that unlike GAN that mimics the Jensen-Shannon divergence and hence its inner maximization is upper bounded by $\log(2)$, WGAN mimics the Wasserstein distance and the inner maximization is unbounded from above. Intuitively, such unboundedness makes the WGAN critic so strong that the WGAN generator (the synthesized transition) cannot find a way out and get stuck at a local optimum. Thereby, we have to modify the WGAN objective to alleviate such a situation. To ensure the boundedness, for a fixed $\delta > 0$, we introduce cut-offs at the WGAN objective so that the inner maximization is upper bounded by $2\delta$:

$$\min_{T'} \max_{\|f\|_{\mathrm{Lip}}\leq 1} \mathbb{E}_{\substack{(s,a)\sim\rho_T^{\alpha,\pi} \\ s'\sim T(\cdot|s,a)}} [\min(\delta, f(s,a,s'))] + \mathbb{E}_{\substack{(s,a)\sim\rho_{T'}^{\alpha,\pi} \\ s'\sim T'(\cdot|s,a)}} [\min(\delta, -f(s,a,s'))]. \tag{6}$$

As $\delta \to \infty$, Eq. 6 recovers the WGAN objective, Eq. 5. Therefore, this is a truncated version of WGAN. To comprehend Eq. 6 further, notice that it is equivalent to

$$\min_{T'} \max_{\|f\|_{\mathrm{Lip}}\leq 1} \mathbb{E}_{\substack{(s,a)\sim\rho_T^{\alpha,\pi} \\ s'\sim T(\cdot|s,a)}} [\min(0, f(s,a,s') - \delta)] + \mathbb{E}_{\substack{(s,a)\sim\rho_{T'}^{\alpha,\pi} \\ s'\sim T'(\cdot|s,a)}} [\min(0, -f(s,a,s') - \delta)]$$

$$\Leftrightarrow \min_{T'} \min_{\|f\|_{\mathrm{Lip}}\leq 1} \mathbb{E}_{\substack{(s,a)\sim\rho_T^{\alpha,\pi} \\ s'\sim T(\cdot|s,a)}} [\max(0, \delta - f(s,a,s'))] + \mathbb{E}_{\substack{(s,a)\sim\rho_{T'}^{\alpha,\pi} \\ s'\sim T'(\cdot|s,a)}} [\max(0, \delta + f(s,a,s'))], \tag{7}$$

which is a hinge loss version of the generative adversarial objective. Such WGAN is introduced in Lim & Ye (2017), where the consistency result is provided and further experiments are evaluated in Zhang et al. (2018). According to Lim & Ye (2017), the inner minimization can be interpreted as the soft-margin SVM. Consequently, it provides a geometric intuition of maximizing margin, which potentially enhances robustness. Finally, because the objective of transition learning is to maximize the cumulative pseudo rewards on the MDP, $T'$ does not directly optimize Eq. 7. Note that the truncation only takes part in the inner minimization:

$$\min_{\|f\|_{\mathrm{Lip}}\leq 1} \mathbb{E}_{\substack{(s,a)\sim\rho_T^{\alpha,\pi} \\ s'\sim T(\cdot|s,a)}} [\max(0, \delta - f(s,a,s'))] + \mathbb{E}_{\substack{(s,a)\sim\rho_{T'}^{\alpha,\pi} \\ s'\sim T'(\cdot|s,a)}} [\max(0, \delta + f(s,a,s'))], \tag{8}$$

which gives us a WGAN critic $f(s,a,s')$. As mentioned, $f$ will be the pseudo reward function. Later, we will introduce a transition learning version of PPO (Schulman et al., 2017) to optimize the cumulative pseudo reward.

---

**Algorithm 1** Model Imitation for Model-Based Reinforcement Learning

1: Parameterize policy $\pi$, $T$, and WGAN critic $f$ with $\theta$, $\phi$, and $w$ respectively. Initialize an empty environment dataset $\mathcal{D}_{\mathrm{env}}$
2: **for** $i = 0, 1, 2, ...$ **do**
3:     Take actions in real environment according to $\pi_\theta$; $\mathcal{D}_{\mathrm{env}} \leftarrow \mathcal{D}_{\mathrm{env}} \cup \mathcal{D}_i$
4:     Pre-train $T_\phi$ and $f_w$ by optimizing Eq. 8 and 11 with $\mathcal{D}_i$ and $\mathcal{D}_{\mathrm{env}}$
5:     **for** $N$ epochs **do**
6:         **for** $n_{\mathrm{transition}}$ epochs **do**
7:             optimize Eq. 8 and 11 over $\phi$ and $w$ with $\mathcal{D}_i$
8:         **end for**
9:         **for** $n_{\mathrm{policy}}$ epochs **do**
10:          update $\pi_\theta$ by TRPO on the data generated by $T_\phi$
11:         **end for**
12:     **end for**
13: **end for**

---

After modifying the WGAN objective, to include both the stochastic and (approximately) deterministic scenarios, the synthesized transition is modeled by a Gaussian distribution $T'(s'|s,a) = T_\phi(s'|s,a) \sim \mathcal{N}(\mu_\phi(s,a), \Sigma_\phi(s,a))$. Although the underlying transitions of tasks like MuJoCo (Todorov et al., 2012) are deterministic, modeling by a Gaussian does not harm the transition learning empirically.

Recall that the synthesized transition is trained on an MDP whose reward function is the critic of the truncated WGAN. To achieve this goal with proper stability, we employ PPO (Schulman et al.,

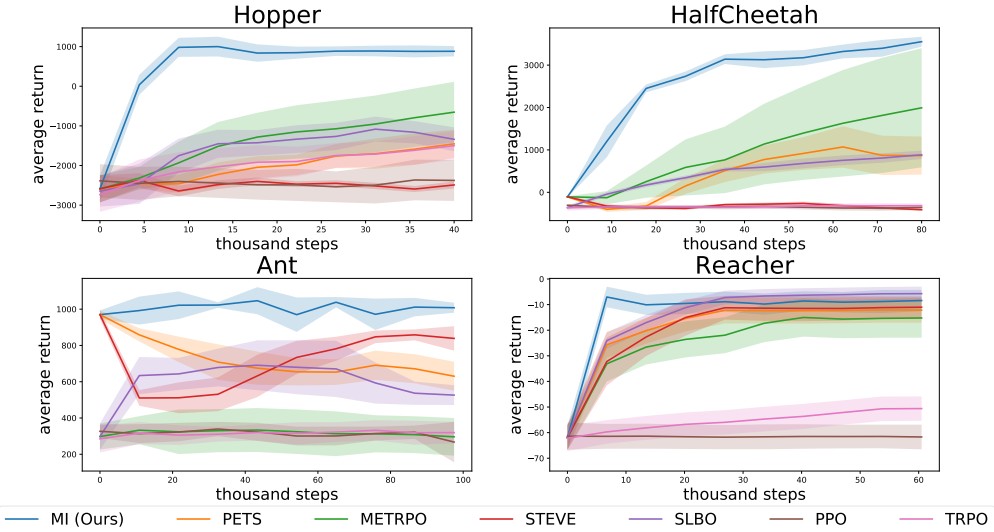

Figure 2: Learning curves of our MI versus two model-free and four model-based baselines. The solid lines indicate the mean of five trials and the shaded regions suggest standard deviation.

2017), which is an efficient approximation of TRPO (Schulman et al., 2015). Note that although the PPO is originally designed for policy optimization, it can be adapted to transition learning with a fixed sampling policy and the PPO objective (Eq. 7 of Schulman et al. (2017))

$$\mathcal{L}_{\text{PPO}}(\phi) = \hat{\mathbb{E}}_t \Big[ \min(r_t(\phi)\hat{A}_t, \ \text{clip}(r_t(\phi), 1 - \epsilon, 1 + \epsilon)\hat{A}_t) \Big], \tag{9}$$

where

$$r_t(\phi) = \frac{T_\phi(s_{t+1}|s_t, a_t)}{T_{\phi_{\text{old}}}(s_{t+1}|s_t, a_t)}, \quad \hat{A}_t : \text{ advantage func. derived from the pseudo reward } f(s_t, a_t, s_{t+1}). \tag{10}$$

To enhance stability of the transition learning, in addition to PPO, we also optimize maximum likelihood, which can be regarded as a regularization. We empirically observe that jointly optimizing both maximum likelihood and the PPO objective attains better transition model for policy gradient. The overall loss of the transition learning becomes

$$\mathcal{L}_{\text{transition}} = -\mathcal{L}_{\text{PPO}} + \alpha \mathcal{L}_{\text{mle}}, \tag{11}$$

where $\mathcal{L}_{\text{mle}}$ is the loss of MLE, which is policy-agnostic and can be estimated with all collected real transitions. For more implementation details, please see Appendix B.1.

We consider a training procedure similar to SLBO (Luo et al., 2019), where they consider the fact that the value function is dependent on the varying transition model. As a result, unlike most of the MBRL methods that have only one pair of model-policy update for each real environment sampling, SLBO proposes to take multiple update pairs for each real environment sampling. Our proposed *model imitation (MI)* method is summarized in Algorithm 1.

## 6 EXPERIMENTS

In the experiment section, we would like to answer the following questions. (1) Does the proposed *model imitation* outperforms the state-of-the-art in terms of sample complexity and average return? (2) Does the proposed *model imitation* benefit from distribution matching and is superior to its model-free and model-based counterparts, TRPO and SLBO?

To fairly compare algorithms and enhance reproducibility, we adopt open-sourced environments released along with a model-based benchmark paper (Wang et al., 2019), which is based on a physical simulation engine, MuJoCo (Todorov et al., 2012). Specifically, we evaluate the proposed algorithm MI on four continuous control tasks including Hopper, HalfCheetah, Ant, and Reacher. For hyperparameters mentioned in Algorithm 1 and coefficients such as entropy regularization $\lambda$, please refer to Appendix B.2.

Table 1: Proportion of bench-marked RL methods that are inferior to MI in terms of $5\%$ t-test. $x/y$ indicates that among $y$ approaches, MI is significantly better than $x$ approaches. The detailed performance can be found in Table 1 of Wang et al. (2019). It should be noted that the reported results in Wang et al. (2019) are the final performance after 200k time-steps whereas ours are no more than 100k time-steps.

|      | Hopper | HalfCheetah | Ant  | Reacher |
|------|--------|-------------|------|---------|
| MBRL | 8/10   | 10/10       | 8/10 | 8/10    |
| MFRL | 3/4    | 2/4         | 4/4  | 3/4     |

We compare to two model-free algorithms, TRPO (Schulman et al., 2015) and PPO (Schulman et al., 2017), to assess the benefit of utilizing the proposed model imitation since our MI (Algorithm 1) uses TRPO for policy gradient to update the agent policy. We also compare MI to four model-based methods. SLBO (Luo et al., 2019) gives theoretical guarantees of monotonic improvement for model-based deep RL and proposes to update a joint model-policy objective. PETS (Chua et al., 2018) propose to employ uncertainty-aware dynamic models with sampling-based uncertainty to capture both aleatoric and epistemic uncertainty. METRPO (Kurutach et al., 2018) shows that insufficient data may cause instability and propose to use an ensemble of models to regularize the learning process. STEVE (Buckman et al., 2018) dynamically interpolates among model rollouts of various horizon lengths and favors those whose estimates have lower error.

Figure 2 shows the learning curves for all methods. In Hopper, HalfCheetah, and Ant, MI converges fairly fast and learns a policy significantly better than competitors'. In Ant, even though MI does not improve the performance too much from the initial one, the fact that it maintains the average return at around 1,000 indicates that MI can capture a better transition than other methods do with only 5,000 transition data. Even though we do not employ an ensemble of models, the curves show that our learning does not suffer from high variance. In fact, the performance shown in Figure 2 indicates that the variance of MI is lower than that of methods incorporating ensembles such as METRPO and PETS.

The questions raised at the beginning of this section can now be answered. The learned model enables TRPO to explore the world without directly access real transitions and therefore TRPO equipped with MI needs much fewer interactions with the real world to learn a good policy. Even though MI is based on the training framework proposed in SLBO, the additional distribution matching component allows the synthesized model to generate similar rollouts to that of the real environments, which empirically gives superior performance because we rely on long rollouts to estimate policy gradient.

To better understand the performance presented in Figure 2, we further compare MI with bench-marked RL algorithms recorded in Wang et al. (2019) including state-of-the-art MFRL methods such as TD3 (Fujimoto et al., 2018) and SAC (Haarnoja et al., 2018). It should be noted that the reported results of Wang et al. (2019) are the final performance after 200k time-steps but we only use up to 100k time-steps to train MI. Table 1 indicates that MI significantly outperforms most of the MBRL and MFRL methods with $50\%$ fewer samples, which verifies that MI is more sample-efficient by incorporating distribution matching.

## 7  CONCLUSION

We have pointed out that the state-of-the-art methods concentrate on learning synthesized models in a supervised fashion, which does not guarantee that the policy is able to reproduce a similar trajectory in the learned model and therefore the model may not be accurate enough to estimate long rollouts. We have proposed to incorporate WGAN to achieve occupancy measure matching between the real transition and the synthesized model and theoretically shown that matching indicates the closeness in cumulative rewards between the synthesized model and the real environment.

To enable stable training across WGANs, we have suggested using a truncated version of WGAN to prevent training from getting stuck at local optimums. The empirical property of WGAN application such as imitation learning indicates its potential to learn the transition with fewer samples than supervised learning. We have confirmed it experimentally by further showing that MI converges much faster and obtains better policy than state-of-the-art model-based and model-free algorithms.

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

## A  PROOFS

### A.1  PROOF FOR WGAN

**Proposition 1** (Consistency for WGAN). *Let $\alpha(s), \pi(a|s), T'(s'|s,a)$ be initial state distribution, policy and synthesized transition. Let $T$ be the true transition, $p(s,a,s') = \rho_T^{\alpha,\pi}(s,a)T(s'|s,a)$ be the discounted distribution of the triple $(s,a,s')$ under the true transition. If the WGAN is trained to its optimal point, we have*

$$T(s'|s,a) = T'(s'|s,a), \; \forall (s,a) \in Supp(\rho_T^{\alpha,\pi}).$$

*Proof.* Because the loss function of WGAN is the 1-Wasserstein distance, we know $p(s,a,s') = p'(s,a,s')$ at its optimal points. Plug in to the Bellman flow constraint Eq. (3),

$$\rho_{T'}^{\alpha,\pi}(s,a) = \pi(a|s)\Big[(1-\gamma)\alpha(s) + \gamma \int \rho_{T'}^{\alpha,\pi}(s',a')T'(s|s',a')ds'da'\Big]$$

$$= \pi(a|s)\Big[(1-\gamma)\alpha(s) + \gamma \int p'(s',a',s)ds'da'\Big]$$

$$\overset{p=p'}{=} \pi(a|s)\Big[(1-\gamma)\alpha(s) + \gamma \int p(s',a',s)ds'da'\Big] = \rho_T^{\alpha,\pi}(s,a)$$

That is,

$$\text{WGAN is opt.} \Leftrightarrow p(s,a,s') = p'(s,a,s') \overset{\text{Bellman}}{\Rightarrow} \rho_T^{\alpha,\pi}(s,a) = \rho_{T'}^{\alpha,\pi}(s,a).$$

Finally, recall $p(s,a,s') \triangleq \rho_T^{\alpha,\pi}(s,a)T(s'|s,a)$ and $p'(s,a,s') \triangleq \rho_{T'}^{\alpha,\pi}(s,a)T'(s'|s,a)$, we arrive at

$$\text{WGAN is opt.} \Rightarrow T(s'|s,a) = T'(s'|s,a), \; \forall (s,a) \in \text{Supp}(\rho_T^{\alpha,\pi}).$$

□

**Theorem 1** (Two-sided Errors for WGAN). *Let $\rho_T^{\alpha,\pi}(s,a)$, $\rho_{T'}^{\alpha,\pi}(s,a)$ be normalized occupancy measures generated by the true transition $T$ and the synthesized one $T'$. Suppose the reward is $L_r$-Lipschitz. If the training error of WGAN is $\epsilon$, then $|R(\pi,T) - R(\pi,T')| \le \epsilon L_r/(1-\gamma)$.*

*Proof.* Observe that the occupancy measure $\rho_T^{\alpha,\pi}(s,a)$ is a marginal distribution of $p(s,a,s') = \rho_T^{\alpha,\pi}(s,a)T(s'|s,a)$. Because the distance between the marginal is upper bounded by that of the joint, we have

$$W_1(\rho_T^{\alpha,\pi}(s,a)||\rho_{T'}^{\alpha,\pi}(s,a)) \le W_1(p(s,a,s')||p'(s,a,s')) = \epsilon,$$

where $W_1$ is the 1-Wasserstein distance. Then, the cumulative reward is bounded by

$$R(\pi,T) = \frac{1}{1-\gamma}\int r(s,a)\rho_T^{\alpha,\pi}(s,a)dsda = R(\pi,T') + \frac{1}{1-\gamma}\int r(s,a)\big(\rho_T^{\alpha,\pi}(s,a) - \rho_{T'}^{\alpha,\pi}(s,a)\big)dsda$$

$$= R(\pi,T') + \frac{L_r}{1-\gamma}\int \frac{r(s,a)}{L_r}\big(\rho_T^{\alpha,\pi}(s,a) - \rho_{T'}^{\alpha,\pi}(s,a)\big)dsda$$

$$\le R(\pi,T') + \frac{L_r}{1-\gamma}\sup_{\|f\|_{\text{Lip}}\le 1}\int f(s,a)\big(\rho_T^{\alpha,\pi}(s,a) - \rho_{T'}^{\alpha,\pi}(s,a)\big)dsda$$

$$= R(\pi,T') + \frac{L_r}{1-\gamma}\sup_{\|f\|_{\text{Lip}}\le 1}\mathbb{E}_{(s,a)\sim\rho_T^{\pi,\alpha}}[f(s,a)] - \mathbb{E}_{(s,a)\sim\rho_{T'}^{\pi,\alpha}}[f(s,a)]$$

$$= R(\pi,T') + \frac{L_r}{1-\gamma}W_1(\rho_T^{\pi,\alpha}||\rho_{T'}^{\pi,\alpha}) \le R(\pi,T') + \epsilon\frac{L_r}{1-\gamma},$$

where the first inequality holds because $r(s,a)/L_r$ is 1-Lipschitz and the last equality follows from Kantorovich-Rubinstein duality Villani (2008). Since $W_1$ distance is symmetric, the same conclusion holds if we interchange $T$ and $T'$, so we arrive at

$$|R(\pi,T) - R(\pi,T')| \le \epsilon L_r/(1-\gamma).$$

□

## A.2 LEMMAS FOR SAMPLING TECHNIQUES

**Lemma 1.** *Let $L \sim Geometric(1 - \beta)$. If $d \geq 1$, then $\mathbb{E}[L^{-1/d}] = O((1 - \beta)^{1/d}/\beta)$.*

*Proof.*

$$\mathbb{E}[L^{-1/d}] = \sum_{i=1}^{\infty} i^{-1/d}(1-\beta)\beta^{i-1} = \frac{1-\beta}{\beta}\sum_{i=1}^{\infty}\frac{\beta^i}{i^{1/d}} = \frac{1-\beta}{\beta}\mathrm{Li}_{1/d}(\beta),$$

where Li is the polylogarithm function. From Wood (1992), the limiting behavior of it is

$$\mathrm{Li}_{1/d}(e^{-\mu}) = \Gamma(1 - 1/d)\mu^{1/d-1}, \text{ as } \mu \to 0^+,$$

where $\Gamma$ is the gamma function. Since $e^{-\mu} \to 1 - \mu$ when $\mu \to 0^+$, we know when $\beta \to 1^-$, $\mathrm{Li}_{1/d}(\beta) \to \Gamma(1-1/d)(1-\beta)^{1/d-1}$. Finally, since $\Gamma(1-1/d) \leq 1$, we conclude that $\mathbb{E}[L^{-1/d}] = O((1-\beta)^{1/d}/\beta)$. □

**Lemma 2.** *Let $\rho_T(s, a)$ be a the normalized occupancy measure generated by the triple $(\alpha, \pi, T)$ with discount factor $\gamma$. Let $\rho_T^\beta(s, a)$ be the normalized occupancy measure generated by the triple $(\rho_T, \pi, T)$ with discount factor $\beta$. If $\gamma > \beta$, then $D_{TV}(\rho_T || \rho_T^\beta) \leq (1 - \gamma)\beta/(\gamma - \beta)$.*

*Proof.* By definition of the occupancy measure we have

$$\rho_T(s, a) = \sum_{i=0}^{\infty}(1 - \gamma)\gamma^i f_i(s, a).$$

$$\rho_T^\beta(s, a) = \sum_{i=0}^{\infty}\sum_{j=0}^{i}(1 - \gamma)\gamma^{i-j}(1 - \beta)\beta^j f_i(s, a),$$

where $f_i(s, a)$ is the density of $(s, a)$ at time $i$ if generated by the triple $(\alpha, \pi, T)$. The TV distance is bounded by

$$D_{TV}(\rho_T || \rho_T^\beta) \leq \frac{1}{2}\sum_{i=0}^{\infty}\left|(1-\gamma)\gamma^i - \sum_{j=0}^{i}(1-\gamma)\gamma^{i-j}(1-\beta)\beta^j\right| = \frac{1}{2}\sum_{i=0}^{\infty}(1-\gamma)\gamma^i\left|1 - \sum_{j=0}^{i}(1-\beta)\left(\frac{\beta}{\gamma}\right)^j\right|$$

$$= \frac{1}{2}\sum_{i=0}^{\infty}(1-\gamma)\gamma^i\frac{1}{\gamma-\beta}\left|-\beta(1-\gamma) + \left(\frac{\beta}{\gamma}\right)^{i+1}(1-\beta)\gamma\right|$$

$$\stackrel{(*)}{=} \frac{(1-\gamma)\beta}{\gamma-\beta}\sum_{i=0}^{M-1} -(1-\gamma)\gamma^i + (1-\beta)\beta^i = \frac{(1-\gamma)\beta}{\gamma-\beta}(\gamma^M - \beta^M)$$

$$\leq \frac{(1-\gamma)\beta}{\gamma-\beta}.$$

where $(*)$ comes from that $-\beta(1 - \gamma) + (\frac{\beta}{\gamma})^i(1 - \beta)\gamma$ is a strictly decreasing function in $i$. Since $\gamma > \beta$, its sign flips from $+$ to $-$ at some index; say $M$. Finally, the sum of the absolute value are the same from $\sum_{i=0}^{M-1}$ and from $\sum_{i=M}^{\infty}$ because the total probability is conservative, and the difference on one side is the same as that on the other. □

## B EXPERIMENTS

### B.1 IMPLEMENTATION DETAILS

We normalize states according to the statistics derived from the first batch of states from the real world. To ensure stability, we maintain the same mean $\mu_0$ and standard deviation $\sigma_0$ throughout the training process.

Instead of directly predicting the next state, we estimate the state difference $s_{t+1} - s_t$ (Kurutach et al., 2018; Luo et al., 2019). Since we incorporate state normalization, the transition network is trained to output $(s_{t+1} - s_t - \mu_0)/\sigma_0$.

To enhance state exploration, we sample real transitions according to policy $\beta \sim \mathcal{N}(\mu_\theta(s), \sigma)$, where $\mu(s)$ is the mean of our Gaussian parameterized policy $\pi_\theta$ and $\sigma$ is a fixed standard deviation. In addition, since model the transition as a Gaussian distribution, we found that matching $\rho_{T'}^{\alpha, \pi_\theta}$ with $\rho_{T}^{\alpha, \beta}$ is empirically more stable and more sample-efficient than matching $\rho_{T'}^{\alpha, \beta}$ with $\rho_{T}^{\alpha, \beta}$.

For policy update, it is shown that using the mean $\mu_\phi$ of the Gaussian-parameterized transition can accelerate policy optimization and better balance exploration and exploitation. In order to enforce the Lipschitz constraint to the WGAN critic $f$, we employ gradient penalty (Gulrajani et al., 2017) with weight 10.

## B.2 HYPERPARAMETERS

|  | HalfCheetah | Hopper | Reacher | Ant |
|---|---|---|---|---|
| $N$ | 10 | | | |
| $\alpha$ | 1 | | | 10 |
| $n_{\text{transition}}$ | 100 | | | |
| $n_{\text{policy}}$ | 20 | 60 | 100 | 30 |
| horizon for model update | 20 | | 10 | 30 |
| entropy regularization | 0.001 | | 0.005 | |

Table 2: List of hyper-parameters adopted in our experiments.

