# OpenReview forum: "Model Imitation for Model-Based Reinforcement Learning"
_ICLR.cc/2020/Conference — Reject_

### Official Review · AnonReviewer1 · 2019-10-21
**Official Blind Review #1**

**Rating:** 6

**Review:**

Review for "Model Imitation for Model-Based Reinforcement Learning".

The paper proposes a type of model-based RL that relies on matching the  distribution of (s,a,s') tuples rather  than using supervised learning to learn an autoregressive model using supervised learning.

I vote to reject the paper for three reasons.

1. The motivation for matching distributions as opposed to learning the model the traditional autoregressive way is lacking. In particular, consider the table lookup case with discrete states and a single actions. Learning the model in this case corresponds to learning a stochastic matrix / Markov Chain. Call this chain P. Define a diagonal matrix whose diagonal contains the ergodic distribution of the chain \Xi. Your framework corresponds to learning a matrix \Xi P, while standard autoregressive models would just learn P. Knowing one gives information about the other - you can go from \Xi P to by normalizing the rows and go from P to \Xi P by computing the stationary distributions. On the other hand, you seem to claim in Figure 1 and in the introduction that your framework is qualitatively different from standard autoregressive models, but the above analysis suggests you are simply approximating a slightly different object, without much of an argument about why this is preferable.

2. The theory section seems a bit underwhelming. In particular:
- Proposition 1 says that we will learn a perfect model given infinite data. That is true, but I am not sure how it helps motivate the paper.
- The presentation of Theorem 1 makes it unclear. In particular, in equation 1, you define R (the return) to depend on the transition model and the policy, but in Theorem 1, you seem to suggest that there is no dependence on the policy.

3. In the experimental section, the Ant plot shows no learning for your method (MI). MI performs well when initialized and does not seem to learn anything (the curve is flat). Please justify why this happens.

I will re-evaluate the paper if the above doubts are cleared up during the revision phase.

Minor point:
Please have the paper proof-read. If you can't find help, please run it through an automated grammar checker. The current version has severe writing problems, which make the writing unclear. Examples:
"we analogize transition learning"
"For deterministic transition, it (what?) is usually optimised with l2-based error"


**Experience Assessment:**

I have published in this field for several years.

**Review Assessment: Checking Correctness Of Derivations And Theory:**

I assessed the sensibility of the derivations and theory.

**Review Assessment: Checking Correctness Of Experiments:**

I assessed the sensibility of the experiments.

**Review Assessment: Thoroughness In Paper Reading:**

I read the paper thoroughly.

---

> ### Author Response · Authors · 2019-11-13
> **Response to Reviewer #1**
>
> Thank you for your insightful comments and suggestions! We have thoroughly checked the paper and corrected the grammatical mistakes. We will continue to polish it to make sure this paper is easy to follow. We would appreciate it if the reviewer could take another look at our changes, and let us know if they would like to revise their score or request additional changes that would alleviate their concerns.
>
> (Q1) The motivation for matching distributions as opposed to learning the model the traditional autoregressive way is lacking. In particular, consider the table lookup case with discrete states and a single action.
> (A1) Your concern is that in some special cases where there is only one action, the transition learning (learning the conditional distribution p(s’|s) ) seems to be equivalent to distribution matching (learning the joint distribution p(s,s’) ). The equivalence is evident when the state space is discrete, as shown in your comment. Here, we argue that there exist major differences under continuous state space and singleton action. In such case, when autoregression/supervised learning is used to learn the conditional distribution p(s’|s), the error in cumulative reward is $O(\epsilon H^2)$ where $H$ is the horizon and $\epsilon$ is the error of $p(s’|s)$, due to the error propagation ($1+2+...+H=O(H^2)$). On the other hand, when the distribution matching is applied, the error is $O(\epsilon H)$, as a special case of Theorem 1. Thus, learning the conditional distribution is hampered by error propagation as the trajectory digresses at least quadratically in the horizon (planning horizon dilemma). In fact, the issue is more severe when the action space is continuous instead of a singleton. In comparison, by Theorem 1, our distribution matching remains resilient under continuous state and action spaces and this is one of the main contributions of this paper. To sum up, the problem of interest (continuous state and action spaces) is highly nontrivial. The traditional ways have theoretical difficulties, and this motivates us to consider the distribution matching proposed in this work.
>
>
> (Q2) The theory section seems a bit underwhelming. In particular:
> 1. Proposition 1 says that we will learn a perfect model given infinite data. That is true, but I am not sure how it helps motivate the paper.
> 2. The presentation of Theorem 1 makes it unclear. In particular, in equation 1, you define R (the return) to depend on the transition model and the policy, but in Theorem 1, you seem to suggest that there is no dependence on the policy.
> (A2)
> 1. Proposition 1 is a sanity check rather than a major conclusion. It tells the optimal point is what we want. The real motivation to apply distribution matching is Theorem 1, because the error in cumulative rewards is more resilient to the horizon, and this mitigates the planning horizon dilemma mentioned above.
> 2. We apologize for the confusion in notation. In Theorem 1, for notational simplicity, we use $R(T)$ instead of $R(\pi,T)$ because the policy is fixed during the distribution matching. A better notation would be $R(\pi,T)$ and $R(\pi,T’)$.
>
>
> (Q3) In the experimental section, the Ant plot shows no learning for your method (MI). MI performs well when initialized and does not seem to learn anything (the curve is flat). Please justify why this happens.
> (A3) We also observed this phenomenon and it should be greatly attributed to the initialization of the policy network. Across all tasks, our method adopts policy initialization with 0.2 standard deviation for actions. We are also surprised that the value is small enough for the Ant agent not to flip over and prevents it from receiving penalties. To fully reproduce the performance of the competitors, we used the original codec as long as it is available on GitHub and therefore we did not share the same initialization across policy networks of different methods. It is interesting that even STEVE and PETS have initializations that allow them to achieve similar scores to ours, but their learned models are unable to capture the dynamic at the beginning, which results in the decrease in the average return.

---

### Official Review · AnonReviewer3 · 2019-10-23
**Official Blind Review #2**

**Rating:** 6

**Review:**

This paper considers the model-based reinforcement learning (MBRL)
problem in which one of the main parts is to learn the transition
probability matrix of the underlying MDP (called Transition Learning -
TL). The motivation is that if the transition model can be learnt from
some real-world trajectories, the agent can improve by interacting with
simulated environment built from the learnt transition model; hence,
the overall number of real-world interactions is reduced.


Given a policy $\pi(a|s)$, and a transition $T(s’|s,a)$, the occupancy
measure $\rho(s,a)$, defined in Eq (2) in the paper, can be
interpreted as the discounted distribution of state-action pairs in
the rollouts. The main idea is to use Wasserstein-GAN (WGAN) to match
the distribution of $p(s,a,s’) = \rho(s,a) T(s’|s,a)$. In particular,
given a policy, the idea is to get some real-world trajectories and
feed into the WGAN framework in which the generator tries to generate
synthesized trajectories and the discriminator tries to discriminate
between them. These trajectories are then used in policy optimization
to obtain a new policy.



The authors presented two theoretical results: 1) Consistency for
WGAN: if WGAN is trained optimally, then the synthesized transition is
the same as the real transition; and 2) Error bound for WGAN: the
difference between cumulative reward under synthesized transition and
that under real transition is upper bounded by a linear function of
WGAN training error. On the experimental side, the authors used the
model-based benchmark library (MBBL, Wang et al. 2019) to compare the
proposed algorithm with several existing algorithms on four MuJoCo
tasks: Hopper, HalfCheetah, Ant, and Reacher.



Overall, I think this work is positive. The theoretical part is
technically sound (there might be an error in the proof of Proposition
1 but it is fixable). The experimental part is also sensible, although
it would be good to include the performance comparisons for other
tasks in MBBL. Details in the comments below.



Main   comments/suggestions:

- MBBL (Wang et al. 2019) has many other tasks (18 in total, the
  authors only include 4 in this work). It would be good if the
  authors can also include the performance of the proposed algorithm
  with respect to those tasks.


- The claim at the end of Proposition 1’s proof (in side the proof), in my assessment, is not
  established. It is clear from the proof that $p(s,a,s’) = p’(s,a,s’)$ is
  a sufficient condition for $\rho_{T}(s,a) = \rho_{T’}(s,a)$, but I’m
  not convinced that it is also a necessary condition. Note that
  $\rho_{T}(s,a)$ and $\rho_{T’}(s,a)$ are two unique solutions of two
  *different* Bellman equations. It would be good if the authors can
  provide the detailed proof for the necessary condition if it is
  true. Nevertheless, the statement of Proposition 1 only claims the
  sufficient condition so this is fixable.





Other minor comments:

- End of Section 1, page 2: the notations $T$, $T’$, $R(T)$, $R(T’)$ are used but not introduced until Section 3.

- Typo, page 2, last paragraph: “IfD method” ==> “LfD” method?

- Algorithm 1, page 6: what are the $\phi$ and $w$ parameters? It looks to me that they are taken from WGAN paper but with no explanation.

- Two references “Syed et al. (2008a)” and “Syed et al. (2008b)” are actually the same.

- Reference “Langlois et al. (2019)” should be corrected as “Wang et al. (2019)” -- see https://arxiv.org/abs/1907.02057




**Experience Assessment:**

I have read many papers in this area.

**Review Assessment: Checking Correctness Of Derivations And Theory:**

I carefully checked the derivations and theory.

**Review Assessment: Checking Correctness Of Experiments:**

I assessed the sensibility of the experiments.

**Review Assessment: Thoroughness In Paper Reading:**

I read the paper thoroughly.

---

> ### Author Response · Authors · 2019-11-13
> **Response to Reviewer #2**
>
> Thank you for your insightful comments and suggestions! We have thoroughly checked the paper and corrected all the mistakes. We will continue to polish it to make sure this paper is easy to follow. We would appreciate it if the reviewer could take another look at our changes, and let us know if they would like to revise their score or request additional changes that would alleviate their concerns.
>
> (Q1) MBBL (Wang et al. 2019) has many other tasks (18 in total, the authors only include 4 in this work). It would be good if the authors can also include the performance of the proposed algorithm with respect to those tasks.
> (A1) Thank you for your advice. The reason why we did not adopt the other tasks for evaluation is because some of them (Pendulum, CartPole, etc) are too easy for most existing works to learn a good policy without clear differences in the average return and the other tasks (Humanoid, Walker2d) are difficult for all MBRL methods to obtain significant policy improvement within our step limit (100k steps).
>
> (Q2) The claim at the end of Proposition 1’s proof (inside the proof), in my assessment, is not established. It is clear from the proof that $p(s,a,s’)=p’(s,a,s’)$ is a sufficient condition for $\rho_T(s,a)=\rho_T’(s,a)$, but I’m not convinced that it is also a necessary condition. Note that $\rho_T(s,a) and \rho_T’(s,a)$ and are two unique solutions of two *different* Bellman equations. It would be good if the authors can provide detailed proof for the necessary condition if it is true. Nevertheless, the statement of Proposition 1 only claims the sufficient condition so this is fixable.
> (A1) We apologize for the mistake made in Proposition 1. Indeed, the necessary direction cannot be established because $\rho_T(s,a),~\rho_{T’}(s,a)$ have dimensions lower than $p(s,a,s’),~p’(s,a,s’)$, and $\rho_T=\rho_{T’}$ cannot imply $p=p’$. Still, having only the sufficient direction is enough for our purpose because it implies the distribution matching leads to consistency in transitions. Thank you a lot for pointing out the mistake.

---

> > ### Comment · AnonReviewer3 · 2019-11-14
> > **promising, but more datasets would strengthen**
> >
> >
> > It would be good to include results on other datasets, perhaps in the appendix (and/or report briefly), to verify the behavior  in any case (whether the task is too easy or too hard).
> >
> > Please also note the feedback from the other review on paper readability, which impacts overall quality.

---

> > > ### Author Response · Authors · 2019-11-15
> > > **Response to Reviewer #2: Regarding Additional Experiments and Paper Readability**
> > >
> > > Thank you for your response and helpful suggestions! We will append additional experiment results in the next version as soon as possible. For the writing quality, we will keep proofreading the paper to enhance its readability.

---

### Official Review · AnonReviewer2 · 2019-10-23
**Official Blind Review #2**

**Rating:** 6

**Review:**

In model-based reinforcement learning methods, in order to alleviate the compounding error induced in rollout-based planning, the paper proposes a distribution matching method, which should be better than a regular supervised learning approach (i.e. minimizing mean square error). Experiments on continuous control domains are presented to show the algorithm’s advantages. The author compares the proposed method with several well-known baselines. The proposed approach is interesting, however, there are some issues.

1. Missing citations. The author should include discussions regarding model-correction methods. For example, Self-Correcting Models for Model-Based Reinforcement Learning, Combating the Compounding-Error Problem with a Multi-step Model. Also, there is a highly relevant work “Learning Latent Dynamics for Planning from Pixels”, in section 4, methods regarding multi-step prediction issue are introduced.

2. The experimental results do not reflect the key issue the author attempts to resolve. MBRL algorithms are very diverse, there are different ways of learning a model and of using a model. Even though the proposed algorithm seems to be better, it is unclear whether it is due to better model accuracy. I think the author should show model accuracy in a rollout of samples comparing against traditional strategies.

===================
I read the author's response and take a look at other reviews. I reconsidered the contribution of the paper and found that my previous concerns may not be that important. I am fine with accepting the paper.

**Experience Assessment:**

I have read many papers in this area.

**Review Assessment: Checking Correctness Of Derivations And Theory:**

I assessed the sensibility of the derivations and theory.

**Review Assessment: Checking Correctness Of Experiments:**

I assessed the sensibility of the experiments.

**Review Assessment: Thoroughness In Paper Reading:**

I made a quick assessment of this paper.

---

> ### Author Response · Authors · 2019-11-13
> **Response to Reviewer #2**
>
> Thank you for your insightful comments and suggestions! We have thoroughly checked the paper and corrected all the mistakes such as missing citations. We will continue to polish it to make sure this paper is easy to follow. We would appreciate it if the reviewer could take another look at our changes, and let us know if they would like to revise their score or request additional changes that would alleviate their concerns.
>
> (Q1) The experimental results do not reflect the key issue the author attempts to resolve. MBRL algorithms are very diverse, there are different ways of learning a model and of using a model. Even though the proposed algorithm seems to be better, it is unclear whether it is due to better model accuracy. I think the author should show model accuracy in a rollout of samples comparing against traditional strategies.
> (A1) The evaluation of the learned model is one of the most challenging parts of model-based reinforcement learning. It’s hard to directly compare our method with previous works in terms of model accuracy because we optimize different objectives. It is obvious that methods that minimize the MSE loss will have much lower training and testing loss. Empirically, directly minimizing such loss does not show clear evidence that the learned model is helpful for the updated policy and the performance may get worse when we use the model to update our policy for multiple gradient steps. Advanced approaches like SLBO [1] are proposed to ensure that we are optimizing a valid objective for both the model and the policy instead of solely for the model. Even though such methods do not improve accuracy, they still achieve much better results than vanilla algorithms. Indeed, directly evaluating the learned model must be a more straightforward way to show our advantages over other works but it is beyond the scope of this work. To address the model quality problem, we theoretically show that the model given by our method is able to approach the ground truth transition in terms of cumulative reward, which is the objective of reinforcement learning and our experiments further confirm the results.
>
> [1] Luo, Yuping, et al. "Algorithmic Framework for Model-based Deep Reinforcement Learning with Theoretical Guarantees." (2018).

---

### Decision · Program_Chairs · 2019-12-19

**Decision:**

Reject

**Comment:**

This paper addresses challenges in offline model learning, i.e., in the setting where some trajectories are given and can be used for learning a model, which in turn serves to train an RL agent or plan action sequences in simulation. A key issue in this setting is that of compounding errors: as the simulated trajectory deviates from observed data, errors build up, leading to suboptimal performance in the target domain. The paper proposes a distribution matching approach that considers trajectory sequence information and provides theoretical guarantees as well as some promising empirical results.

Several issues were raised by reviewers, including missing references, clarity issues, questions about limitations of the theoretical analysis, and limitations of the empirical validation. Many of the issues raised by reviewers were addressed by the authors during the rebuttal phase.

At the same time, several issues remain. First, the authors committed to adding results for additional tasks (initially deemed too easy or too hard to show differences). Even if the tasks show little separation between methods, these would be important data points to include as they support additional comparisons with prior and future work. The AC has to assess the paper without taking promised additional results into account. Second, questions about the results for Ant are not sufficiently addressed. The plot shows no learning. The author response mentions initialization but this is not deemed a sufficient explanation. Given the remaining questions, my assessment is that the quality and contribution of the submission are not yet ready for publication at the current stage.